# What's new? Summarizing Contributions in Scientific Literature

## Abstract

With thousands of academic articles shared on a daily basis, it has become increasingly difficult to keep up with the latest scientific findings. To overcome this problem, we introduce a new task of *disentangled paper summarization*, which seeks to generate separate summaries for the paper contributions and the context of the work, making it easier to identify the key findings shared in articles. For this purpose, we extend the S2ORC corpus of academic articles, which spans a diverse set of domains ranging from economics to psychology, by adding disentangled "contribution" and "context" reference labels. Together with the dataset, we introduce and analyze three baseline approaches: 1) a unified model controlled by input code prefixes, 2) a model with separate generation heads specialized in generating the disentangled outputs, and 3) a training strategy that guides the model using additional supervision coming from inbound and outbound citations. We also propose a comprehensive automatic evaluation protocol which reports the *relevance*, *novelty*, and *disentanglement* of generated outputs. Through a human study involving expert annotators, we show that in 79%, of cases our new task is considered more helpful than traditional scientific paper summarization.

## 1 Introduction

With the growing popularity of open-access academic article repositories, such as arXiv or bioRxiv, disseminating new research findings has become nearly effortless. Through such services, tens of thousands of scientific papers are shared by the research community every month[1]. At the same time, the unreviewed nature of mentioned repositories and the sheer volume of new publications has made it nearly impossible to identify relevant work and keep up with the latest findings.

Scientific paper summarization, a subtask within automatic text summarization, aims to assist researchers in their work by automatically condensing articles into a short, human-readable form that contains only the most essential information. In recent years, abstractive summarization, an approach where models are trained to generate fluent summaries by paraphrasing the source article, has seen impressive progress. State-of-the-art methods leverage large, pre-trained models (Raffel et al., 2019; Lewis et al., 2020), define task-specific pre-training strategies (Zhang et al., 2019), and scale to long input sequences (Zhao et al., 2020; Zaheer et al., 2020). Available large-scale benchmark datasets, such as arXiv and PubMed (Cohan et al., 2018), were automatically collected from online archives and repurpose paper abstracts as reference summaries. However, the current form of scientific paper summarization where models are trained to generate paper abstracts has two caveats: 1) often, abstracts contain information which is not of primary importance, 2) the vast majority of scientific articles come with human-written abstracts, making the generated summaries superfluous.

To address these shortcomings, we introduce the task of disentangled paper summarization. The new task's goal is to generate two summaries simultaneously, one strictly focused on the summarized article's novelties and contributions, the other introducing the context of the work and previous efforts. In this form, the generated summaries can target the needs of diverse audiences: senior researchers and field-experts who can benefit from reading the summarized contributions, and newcomers who can quickly get up to speed with the intricacies of the addressed problems by reading the context summary and get a perspective of the latest findings from the contribution summary.

---

[1] https://arxiv.org/stats/monthly_submissions

For this task, we introduce a new large-scale dataset by extending the S2ORC (Lo et al., 2020) corpus of scientific papers, which spans multiple scientific domains and offers rich citation-related metadata. We organize and process the data, and extend it with automatically generated contribution and context reference summaries, to enable supervised model training. We also introduce three abstractive baseline approaches: 1) a unified, controllable model manipulated with descriptive control codes (Fan et al., 2018; Keskar et al., 2019), 2) a one-to-many sequence model with a branched decoder for multi-head generation (Luong et al., 2016; Guo et al., 2018), and 3) an information-theoretic training strategy leveraging supervision coming from the citation metadata (Peyrard, 2019). To benchmark our models, we design a comprehensive automatic evaluation protocol that measures performance across three axes: relevance, novelty, and disentanglement. We thoroughly evaluate and analyze the baselines models and investigate the effects of the additional training objective on the model's behavior. To motivate the usefulness of the newly introduced task, we conducted a human study involving human annotators in a hypothetical paper-reviewing setting. The results find disentangled summaries more helpful in 79% of cases in comparison to abstract-oriented outputs. Code, model checkpoints, and data preparation scripts introduced in this work are available at `https://github.com/salesforce/disentangled-sum`.

## 2 RELATED WORK

Recent trends in abstractive text summarization show a shift of focus from designing task-specific architectures trained from scratch (See et al., 2017; Paulus et al., 2018) to leveraging large-scale Transformer-based models pre-trained on vast amounts of data (Liu & Lapata, 2019; Lewis et al., 2020), often in multi-task settings (Raffel et al., 2019). A similar shift can be seen in scientific paper summarization, where state-of-the-art approaches utilize custom pre-training strategies (Zhang et al., 2019) and tackle problems of summarizing long documents (Zhao et al., 2020; Zaheer et al., 2020). Other methods, at a smaller scale, seek to utilize the rich metadata associated with scientific articles and combine them with graph-based methods (Yasunaga et al., 2019). In this work, we combine these two lines of work and propose models that benefit from pre-training procedures, but also take advantage of task-specific metadata.

Popular large-scale benchmark datasets in scientific paper summarization (Cohan et al., 2018) were automatically collected from open-access paper repositories and consider article abstracts as the reference summaries. Other forms of supervision have also been investigated for the task, including author-written highlights (Collins et al., 2017), human annotations and citations (Yasunaga et al., 2019), and transcripts from conference presentations of the articles (Lev et al., 2019). In contrast, we introduce a large-scale automatically collected dataset with more fine-grained references than abstracts, which also offers rich citation-related metadata.

Update summarization (Dang & Owczarzak) defines a setting in a collection of documents with partially overlapping information is summarized, some of which are considered prior knowledge. The goal of the task is to focus the generated summaries on the novel information. Work in this line of research mostly focuses on novelty detection in news articles (Bysani, 2010; Delort & Alfonseca, 2012) and timeline summarization (Martschat & Markert, 2018; Chang et al., 2016) on news and social media domains. Here, we propose a novel task that is analogous to update summarization in that it also requires contrasting the source article with the content of other related articles which are considered pre-existing knowledge.

## 3 TASK

Given a source article $D$, the goal of disentangled paper summarization is to simultaneously summarize the *contribution* $y_{con}$ and *context* $y_{ctx}$ of the source article. Here, contribution refers to the novelties introduced in the article $D$, such as new methods, theories, or resources, while context represents the background of the work $D$, such as a description of the problem or previous work on the topic. The task inherently requires a relative comparison of the article with other related papers to effectively disentangle its novelties from pre-existing knowledge. Therefore, we also consider two sets of citations: inbound citations $C_I$ and outbound citations $C_O$ as potential sources of useful information for contrasting the article $D$ with its broader field. Inbound citations refer to the set of papers that cite $D$, *i.e.* relevant future papers, while outbound citations are the set of papers that

Table 1: Token length statistics on the training split of our dataset compared to existing scientific paper summarization datasets. Contribution summaries tend to be shorter than context summaries.

| Dataset | #Examples | Avg. #Tokens | | | | |
|---|---|---|---|---|---|---|
| | | Paper $D$ | Inbound $C_I$ | Outbound $C_O$ | Contribution $y_{con}$ | Context $y_{ctx}$ |
| ArXiv (Train) | 203037 | 4938 | - | - | 220 (Total summary) | |
| PubMed (Train) | 119924 | 3016 | - | - | 203 (Total summary) | |
| Ours - Train | 805152 | 6351 | 925 | 877 | 136 | 236 |
| Valid | 36129 | 6374 | 922 | 875 | 135 | 236 |
| Test | 54242 | 6350 | 927 | 892 | 136 | 237 |

$D$ cites, *i.e.* relevant previous papers. With its unique set of goals, the task of disentangled paper summarization poses a novel set of challenges for automatic summarization systems to overcome: 1) identifying salient content of $D$ and related papers from $C_I$ and $C_O$, 2) comparing the content of $D$ with each document from the citations, and 3) summarizing the article along the two axes: contributions and context.

## 3.1 DATASET

Current benchmark datasets used for the task of scientific paper summarization, such as arXiv and PubMed (Cohan & Goharian, 2015), are limited in size, the number of domains, and lack of citation metadata. Thus, we construct a new dataset based on the S2ORC (Lo et al., 2020) corpus, which offers a large collection of scientific papers spanning multiple domains along with rich citation-related metadata, such as citation links between papers and annotated citation spans. Specifically, we carefully curate the data available in the S2ORC corpus and extend it with new reference labels.

**Data Curation**  Some papers in the S2ORC corpus[2] do not contain a complete set of information required by our summarization task: paper text, abstract, and citation metadata. We remove such instances and construct a paper summarization dataset in which each example a) has an abstract and body text, and b) has at least 5 or more inbound and outbound citations, $C_I$ and $C_O$ respectively. In cases where a paper has more than $K$ incoming or outgoing citations, we first sample $K$ citations for each of incoming and outgoing citations and sort them in descending order by the number of their respective citation from and to the target paper. $K$ is a hyperparameter and we choose $K = 20$ in this work.

**Citation Span Extraction**  Each article in the set of inbound and outbound citations can be represented by its full text, abstract, or the span of text associated with the citation. In this study, we follow Qazvinian & Radev (2008) and Cohan & Goharian (2015) in representing citations with the sentences in which the citation occurs.[3] Thus, an outbound citation is represented by a sentence from the source paper. Usually, such sentences directly refer to the cited paper and place its content in relation to the source paper. Analogously, an inbound citation is represented by sentences from the citing paper and relates its content with the source paper.

**Reference Generation**  Our approach relies on the availability of reference summaries for both contributions and contexts. However, such annotations are not provided or easily extractable from the S2ORC corpus, and collecting expert annotations is infeasible due to the associated costs. Therefore, we apply a data-driven approach to automatically extract contribution and context reference summaries from the available paper abstracts. First, we manually label 400 abstracts sampled from the training set. Annotations are done on a sentence-level with binary labels indicating *contribution*- and *context*-related sentences.[4] This procedure yields 3341 sentences with associated binary labels, which we refer to as golden standard references. Next, we fine-tune an automatic sentence classifier using the golden standard data. As our classifier we use SciBERT (Beltagy et al., 2019), which after fine-tuning achieves 86.3% accuracy and 0.932 for Area under ROC curve in classifying *contribution* and *context* sentences on a held-out test set. Finally, we apply the fine-tuned classifier

---

[2]Release ID: 20190928.

[3]If a publication is cited multiple times within a source article we concatenate all relevant sentences.

[4]Sentences not labeled as contribution are considered context, we leave finer-grained labels for future work.

Figure 1: Model diagram. Left: CONTROLCODE model, in which inputs are prefixed with a prompt symbol and passed to a shared model to control the output mode. Right: MULTIHEAD model, which shares all of the model's parameters apart from the last decoder layer for different output modes, and chooses the final decoder layer accordingly to control the output mode.

to generate reference labels for all examples in our dataset, which we refer to as silver standard references. The statistics of the resulting dataset are shown in Table 1.

## 4 MODELS

Our goal is to build an abstractive summarization system which has the ability to generate contribution and context summaries based on the source article. To achieve the necessary level of controllability, we propose two independent approaches building on encoder-decoder architectures:

**CONTROLCODE (CC)** A common approach to controlling model-generated text is by conditioning the generation procedure on a control code associated with the desired output. Previous work on controllable generation (Fan et al., 2018; Keskar et al., 2019) showed that prepending a special token or descriptive prompt to the model's input during training and inference is sufficient to achieve fine-grained control over the generated content. Following this line of work, we modify our training instances by prepending textual control codes, `contribution:` or `context:`, to the summarized articles. During training, all model parameters are updated for each data instance and the model is expected to learn to associate the provided prompt with the correct output mode. The approach does not require changes in the architecture, making it straightforward to combine with existing large-scale, pre-trained models. The architecture is shown on the left of Figure 1.

**MULTIHEAD (MH)** An alternative way of controlling generation is by explicitly allocating layers within the model specifically for the desired control aspects. Prior work investigating multi-task models (Luong et al., 2016; Guo et al., 2018) showed the benefits of combining shared and task-specific layers within a single, multi-task architecture. Here, the encoder shares all parameters between the two generation modes, while the decoder shares all parameters, apart from the final layer, which splits into two generation branches. During training, each branch is individually updated with gradients from the associated mode. The model shares the softmax layer weights between the output branches under the assumption that token-level vocabulary distributions are similar in the two generation modes due to the common domain. This approach is presented on the right of Figure 1.

### 4.1 INFORMATIVENESS-GUIDED TRAINING

Peyrard (2019) proposed an information-theoretic perspective on text summarization which decomposes the criteria of a good summary into redundancy, relevance, and informativeness. Among these criteria, *informativeness* measures the user's degree of surprise after reading a summary given their background knowledge, and can be formally defined as:

$$Inf(D, K) = -\sum_i P_D(\omega_i) \log P_K(\omega_i), \tag{1}$$

where $\omega_i$ is a primitive semantic unit, $P_K$ is the probability over the unit under the user's knowledge, $P_D$ is the probability over the unit with respect to the source document, and $i$ is an index over all semantic units within a summary.

As defined by Peyrard (2019), informativeness is in direct correspondence to contribution summarization. Paper contributions are novel contents introduced to the community, which causes surprisal given the general knowledge about the state of the field. Therefore, in this work we explore utilizing this measure as an auxiliary objective that is optimized during training. We define the semantic unit $\omega_i$ as the summary itself[5], which enables a simple interpretation of the corresponding probabilities. We estimate $P_D$ as the likelihood of the summary given the paper content, $P_D(\omega_i) = p(y \,|\, D)$. Since each paper is associated with a unique context and background knowledge, we treat the background knowledge as all relevant papers published before the source paper, *i.e.*, outbound citations $C_O$. Therefore, $P_K$ is estimated as the likelihood of the summary given the previous work, $P_K(\omega_i) = p(y \,|\, C_O)$. We formulate the informativeness function as:

$$Inf(D, K) = \begin{cases} -p(y_{con} \,|\, D) \log p(y_{con} \,|\, C_O) & \text{if generating contributions} \\ -p(y_{ctx} \,|\, D) \log p(y_{ctx} \,|\, C_I) & \text{otherwise} \end{cases}, \qquad (2)$$

where the conditioning depends on the generation mode of the model, and aim to maximize it during the training procedure. The estimation of each term is done by a forward pass on the model with corresponding input and output pairs: $p(y_{con} \,|\, C_O)$ is computed by estimating the probability of $y_{con}$ when feeding $C_O$ as the encoder input.

Combined with a cross entropy loss $L_{CE}$, we obtain the final objective which we aim to the minimize during training:

$$L = L_{CE} - \lambda \, Inf(D, K), \qquad (3)$$

where $\lambda$ is a scaling hyperparameter determined through cross-validation.

## 5 EXPERIMENTS AND RESULTS

In this section, we describe the experimental environment and report automatic evaluation results. We consider four model variants:

- **CC**, **CC+INF**: CONTROLCODE model without and with the informativeness objective,
- **MH**, **MH+INF**: MULTIHEAD model without and with the informativeness objective.

### 5.1 EVALUATION

We perform automatic evaluation of the system outputs $(s_{con}, s_{ctx})$ against the silver standard references $(y_{con}, y_{ctx})$. For this purpose, we have designed a comprehensive evaluation protocol, shown in Figure 2, based on existing metrics that evaluates the performance of models across 3 dimensions:

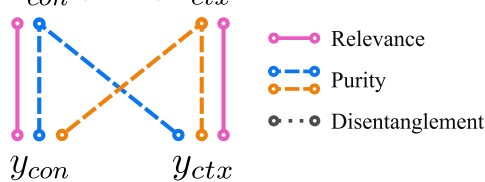

Figure 2: Diagram illustrating the evaluation protocol assessing summaries along 3 axes: relevance, purity, and disentanglement.

**Relevance** Generated summaries should closely correspond with the available reference summaries. We measure the lexical overlap and semantic similarity between $(s_{con}, y_{con})$ and $(s_{ctx}, y_{ctx})$ using ROUGE (R-$i$) (Lin, 2004) and BERTScore (Zhang et al. 2020; BS), respectively.

**Purity** Generated contribution summary should closely correspond with its respective reference summary, but should not overlap with the context reference summary. We measure the lexical overlap between $s_{con}$ and $(y_{con}, y_{ctx})$ using NouveauROUGE $_{con}$ (N$_{con}$-$i$) (Conroy et al., 2011). The metric reports an aggregate score defined as a linear combination between the two components:

$$\text{NouveauROUGE}_{con}\text{-}i = \alpha_0^i + \alpha_1^i \text{ROUGE-}i(s_{con}, y_{con}) + \alpha_2^i \text{ROUGE-}i(s_{con}, y_{ctx}),$$

where weights $\alpha_j^i$ were set by the original authors to favor outputs with maximal and minimal overlap with related and unrelated references, accordingly. Analogously, we calculate N$_{ctx}$-$i$ in

---

[5] For simplicity in modeling, we chose the entire summary. However, this goes against the requirement set by Peyrard (2019) that $\omega_i$ is a *primitive* semantic unit, because a paragraph's meaning can be decomposed into higher granular units.

reverse direction between $s_{ctx}$ and $(y_{ctx}, y_{con})$. Purity P-$i$ is defined as the average novelty in both directions:

$$\text{Purity-}i = (\text{N}_{con}\text{-}i + \text{N}_{ctx}\text{-}i)/2; \quad (\text{P-}i).$$

**Disentanglement** Generated contribution and context summaries should have minimal overlap. We measure the degree of lexical overlap and semantic similarity between $(s_{con}, s_{ctx})$ using ROUGE and BERTScore, respectively. To maintain consistency across metrics (higher is better) we report disentanglement scores as complements of the associated metrics:

$$\text{DisROUGE-}i = 100 - \text{ROUGE-}i; \quad (\text{D-}i),$$
$$\text{DisBERTScore} = 100 - \text{BERTScore}; \quad (\text{DBS}).$$

## 5.2 IMPLEMENTATION DETAILS

Our models build upon distilBART[6] (Sanh et al., 2019; Wolf et al., 2019), a Transformer-based (Vaswani et al., 2017), pre-trained sequence-to-sequence architecture distilled from BART (Lewis et al., 2020). Specifically, we used a model with 6 self-attention layers in both the Encoder and Decoder. Weights were initialized from a model fine-tuned on a news summarization task.[7] For the MULTIHEAD model, the final layer of the decoder was duplicated and initialized with identical weights. We fine-tuned on the training set for 80000 gradient steps with a fixed learning rate of $3.0 \times 10^{-5}$ and choose the best checkpoints in terms of ROUGE-1 scores on the validation set. The loss scaling hyparameter $\lambda$ (Eq. 3) was set to 0.05 and 0.01 for the CONTROLCODE and MULTIHEAD models, accordingly. Input and output lengths were set to 1024 and 200, respectively. At inference time, we decoded using beam search with beam size 5. The evaluation was performed using SummEval toolkit (Fabbri et al., 2020).

## 5.3 RESULTS

In Table 2 we report results from the automatic evaluation protocol described in Subsection 5.1.

**Relevance** Across most models and metrics, relevance scores for context generation are higher than those for contribution summarization. Manual inspection revealed that in some cases generated context summaries also include article contribution information, while this effect was not observed in the reverse situation. Considering that silver standard annotations may contain noisy examples with incorrectly separated references, we suspect that higher ROUGE scores for context summaries may be caused by noisy predictions coinciding with noisy references. Examples of such summaries are shown in the Appendix D. We also observe that informativeness-guided models (+INF) perform on par with their respective base versions, and the additional training objective does not affect the performance on the relevance metric. This insight corroborates with Peyrard (2019) who defines informativeness and relevance as orthogonal criteria.

**Purity** While the informativeness objective was designed to improve the novelty of generated summaries, results show an opposite effect, where informativeness-guided models slightly under-perform their base counterparts. The true reason for such behavior is unknown, however, it might be an indicator that the outbound citations $C_O$ are not a good approximation of reference context summaries $y_{ctx}$, or the relationship between the two is weak. This effect is more evident in the Medical and Biology domains, which are the two most frequent domains in the dataset.

**Disentanglement** Results indicate that CONTROLCODE-based models perform better than MULTIHEAD approaches in terms of generating disentangled outputs. This comes as a surprise given that the CC models share all parameters between the two generation modes, but might indicate that the two tasks contain complementary training signals. We also noticed that, both informativeness-guided models performed better in terms of D-1.

Based on both purity and disentanglement evaluations, we suspect that the informativeness objective does guide the models to output more disentangled summaries (second term in Eq 2), but the signal

---

[6]We did not observe a substantial difference in performance between distilBART and BART.

[7]Model weights are available at `https://huggingface.co/sshleifer/student_cnn_6_6`

Table 2: Automatic evaluation results on the test set. For all metrics, higher values indicate better results. Con and Ctx refer to contribution summary and context summary, respectively. Purity and Disentanglement are measaured on the pairs of contribution and context summaries.

| Model | | Relevance | | | | Purity | | Disentanglement | | | |
|---|---|---|---|---|---|---|---|---|---|---|---|
| | | R-1 | R-2 | R-L | BS | P-1 | P-2 | D-1 | D-2 | D-L | DBS |
| CC | Con | 39.16 | 15.96 | 24.65 | 63.22 | 2.77 | 3.69 | 52.95 | 72.18 | 69.12 | 33.62 |
| | Ctx | 41.84 | 17.24 | 24.55 | 63.78 | | | | | | |
| CC+INF | Con | 38.92 | 15.95 | 24.65 | 62.94 | 2.75 | 3.68 | 53.68 | 71.97 | 68.46 | 34.09 |
| | Ctx | 41.49 | 17.03 | 24.50 | 63.40 | | | | | | |
| MH | Con | 39.20 | 15.98 | 24.72 | 63.04 | 2.73 | 3.68 | 50.89 | 69.51 | 65.97 | 32.51 |
| | Ctx | 41.67 | 17.23 | 24.65 | 63.77 | | | | | | |
| MH+INF | Con | 38.74 | 15.90 | 24.59 | 62.70 | 2.68 | 3.60 | 53.35 | 71.47 | 67.20 | 33.86 |
| | Ctx | 40.39 | 16.31 | 23.83 | 62.85 | | | | | | |

is not strong enough to focus on generating the appropriate content (first term in Eq 2). It is also clear that the MULTIHEAD model benefits more from the additional training objective.

# 6 ANALYSIS

## 6.1 QUALITATIVE ANALYSIS

To better understand the strengths and shortcomings of our models, we performed a qualitative study of model outputs. Table 3 shows an example of generated summaries compared with the original abstract of the summarized article. Our model successfully separates the two generation modes and outputs coherent and easy to follow summaries. The contribution summary clearly lists the novelties of the work, while the context summary introduces the task at hand and explains its importance. In comparison, the original abstract briefly touches on many aspects: the context, methods used, and contributions, but also offers details that are not of primary importance, such as the detailed about the simulation environment.

More generally, the described trends hold across summaries generated by our models. The model outputs are fluent, abstractive, offer good separation between modes, and are on topic. However, the factual correctness of summaries could not be assessed due to the highly specialized content and language of the summarized articles. An artifact noticed in a few instances of the inspected outputs was leakage of contribution information into context summaries. Other examples of generated summaries are included in the Appendix D.

## 6.2 PER-DOMAIN PERFORMANCE

Taking advantage of the rich metadata associated with the S2ORC corpus, we analyze the performance of models across the 10 most frequent scientific domains. Table 4 shows the results of contribution summarization using the CONTROLCODE[8] model. While ROUGE-1 scores oscillate around 40 points for most academic fields, the results indicate that summarizing documents from the Medical domain is particularly difficult, with models scoring about 7 points below average. Manual inspection of instances with low scores (R-1 < 20), exposed that contribution summaries in the Medical domain are highly quantitative (*e.g.* "Among these

Table 4: Relevance evaluation of contribution summaries for the top 10 domains generated using the CONTROLCODE model. Performance on Medicine domain is paricularly low.

| Metric | R-1 | R-2 | R-L | BS |
|---|---|---|---|---|
| Biology | 40.63 | 17.01 | 25.59 | 64.23 |
| Medicine | 33.97 | 13.08 | 21.73 | 61.75 |
| Mathematics | 40.13 | 15.56 | 24.42 | 61.58 |
| Computer science | 43.54 | 16.41 | 25.86 | 63.43 |
| None | 40.31 | 18.14 | 26.68 | 64.00 |
| Psychology | 39.51 | 15.56 | 24.34 | 62.95 |
| Physics | 40.09 | 15.85 | 24.89 | 62.10 |
| Chemistry | 40.44 | 17.77 | 26.14 | 63.93 |
| Economics | 39.56 | 14.25 | 23.41 | 60.91 |
| Materials science | 42.52 | 18.96 | 27.57 | 65.25 |

treated . . . retinopathy was noted in X%"). While other domains such as Biology also suffer from the same phenomenon, low-scoring quantitative summaries were 1.9 times more frequent in Medicine than in Biology. An investigation into the domain distribution in our dataset (Appendix) revealed

---

[8]The remaining models exhibit the same pattern.

Table 3: Generated samples compared with the original and generated abstracts of the associated paper. The second rows shows the output decoded from DistilBART fine-tuned on our dataset, the third rows shows the outputs from CONTROLCODE model. Our model successfully generates disentangled content, thus making it easier to follow than the abstract.

**Original Abstract:** Energy optimization in buildings by controlling the Heating Ventilation and Air Conditioning (HVAC) system is being researched extensively. In this paper, a model-free actor-critic Reinforcement Learning (RL) controller is designed using a variant of artificial recurrent neural networks called Long-Short-Term Memory (LSTM) networks. Optimization of thermal comfort alongside energy consumption is the goal in tuning this RL controller. The test platform, our office space, is designed using SketchUp. Using OpenStudio, the HVAC system is installed in the office. The control schemes (ideal thermal comfort, a traditional control and the RL control) are implemented in MATLAB. Using the Building Control Virtual Test Bed (BCVTB), the control of the thermostat schedule during each sample time is implemented for the office in EnergyPlus alongside local weather data. Results from training and validation indicate that the RL controller impoves thermal comfort by an average of 15% and energy efficiency by an average of 2.5% as compared to other strategies mentioned.

**Generated Abstract:** Despite the advances in research on HVAC control algorithms, most field equipment is controlled using classical methods that include hysteresis/on/off and Proportional Integral and Derivative (PID) controllers. These classical methods do not perform optimally. The high thermal inertia of buildings induces large time delays in the building dynamics, which cannot be handled efficiently by the simple on/off controllers. However, due to the high non-linearity in building dynamics coupled with uncertainties such as weather, energy pricing, etc., these PID controllers require extensive retuning or auto-tuning capabilities, which increases the difficulty and complexity of the control problem. In this work, we introduce novel control algorithms from a branch of machine learning called reinforcement learning. From a controls perspective, reinforcement learning algorithms can be considered as direct adaptive optimal control. Like optimal control, reinforcement training algorithms minimize the cumulative sum of costs over a time horizon. Unlike traditional optimization algorithms can learn optimal control actions

**Contribution:** In this work, we introduce novel control algorithms from a branch of machine learning called reinforcement learning. In our current approach, the impetus is thermostat control. Instead of traditional on/off heating and cooling control, reinforcement learning is utilized to set this schedule to obtain improved Predicted Mean Vote (PMV)-based thermal comfort at an optimal energy expenditure. Hence, a thermostats schedule is computed using an RL controller. The results show that the Q-learning algorithm can learn to adapt to time-varying and nonlinear system dynamics without explicit identification of the plant model in both systems and controls.
**Context:** The Heating, Ventilation and Air Conditioning (HVAC) systems can account for up to 50% of total building energy demand. In the hopes of moving toward a greener, more energy-efficient future, a significant improvement in energy efficiency is needed to achieve this goal. Despite the advances in research on HVAC control algorithms, most field equipment is controlled using classical methods that include hysteresis/on/off and Proportional Integral and Derivative controllers. However, due to the high nonlinearity in building dynamics coupled with uncertainties such as weather, energy pricing, etc., these PID controllers require extensive retuning or auto-tuning capabilities, which increases the difficulty and complexity of the control problem. The high thermal inertia of buildings induces large time delays in the building dynamics, which cannot be handled efficiently by the simple on/off controllers.

that Biology and Medicine are the two best represented fields in the corpus, with Biology having over twice as many examples. We hypothesize that the poor performance of models stems from the fact that generating such quantitative summaries requires a deeper, domain-specific understanding of the source document and the available in-domain training data is insufficient to achieve that goal.

## 6.3 HUMAN EVALUATION OF USEFULNESS

To assess the usefulness of the newly introduced task to the research community, we conducted a human study involving expert annotators. The study aimed to compare disentangled papers summaries with traditional, abstract-based summaries in a hypothetical paper reviewing setting. Judges were shown both types of summaries side by side and asked to pick one which would be more helpful for conducting the paper review. We show a

Table 5: Usefulness of disentangled summaries in percentage, *e.g.*, Annotator 1 (A1) chose the disentangled summaries 82% out of all the samples from S2ORC.

| Dataset | A1 | A2 | A3 | AVG. |
|---------|-----|-----|-----|------|
| S2ORC | 82% | 78% | 70% | 77% |
| CORD | 88% | 76% | 78% | 81% |

screen shot of the annotation user interface in Fig 3. Abstract-based summaries were generated by a model with a configuration identical to the models previously introduced in this work, trained to generate full abstracts using the same training corpus. Annotators that participated in this study hold graduate degrees in technical fields and are active in the research community, however, they were not involved or familiar with this work prior to this experiment. The study used 100 examples, out of which 50 were decoded on the test split of the adapted S2ORC dataset, while the other 50 were generated in a zero-shot fashion from articles in the CORD dataset (Wang et al., 2020), a recently introduced collection of papers related to COVID-19. The inter-annotator agreement measured by Fleiss' Kappa was $0.41$ and $0.33$ for the S2ORC and CORD datasets, respectively. Results in Table 5 show the proportion of all examples where the annotators preferred the disentangled summaries over the generated abstracts. The numbers indicate a strong preference from the judges for disentangled summaries, in the case of both S2ORC and CORD examples. The values on CORD samples are slightly higher than those on S2ORC; we suspect this being due to the fact that the annotators were

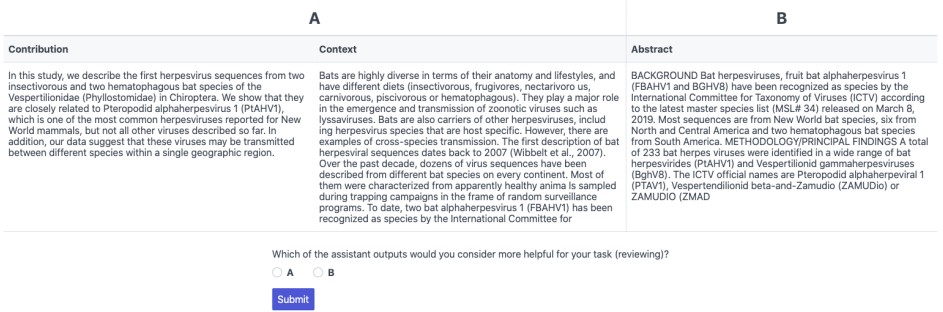

Figure 3: The annotation interface. Summaries indicated with A and B are disentangled summaries and a generated abstract, respectively.

less familiar with the topics described in Covid-related publications and would require more help to review such articles.

## 6.4 EVALUATION AGAINST GOLD ANNOTATIONS

As discussed in Section 3.1, contribution and context labels are assigned automatically using a data-driven classifier, which could introduce errors in the process. Therefore, we created a gold standard evaluation set by manually annotating 100 samples from the test set and report the evaluation results in Table 6. A sharp drop in ROUGE scores for the context summaries is caused by some examples receiving zero scores for generating context summaries when the manual annotators judged that there are not existent. The overall trend of CONTROLCODE model outpeforming MULTIHEAD model is still observed in the evaluation. More importantly, a reverse tendency is noticeable when the two models are trained with the informativeness objective. Specifically, the MULTIHEAD model showed significant improvement in terms of novelty and disentanglement.

Table 6: Automatic evaluation results on 100 samples from the test set manually annotated for contributions. For all metrics, higher values indicate better results.

| Model | | Relevance | | | | Purity | | Disentanglement | | | |
|---|---|---|---|---|---|---|---|---|---|---|---|
| | | R-1 | R-2 | R-L | BS | P-1 | P-2 | D-1 | D-2 | D-L | DBS |
| CC | Con | 39.37 | 15.86 | 24.73 | 63.28 | 2.30 | 3.22 | 52.81 | 71.52 | 68.36 | 33.05 |
| | Ctx | 30.59 | 11.22 | 19.08 | 55.76 | | | | | | |
| CC+INF | Con | 38.38 | 15.21 | 23.47 | 62.59 | 2.17 | 3.10 | 52.49 | 69.64 | 66.60 | 32.76 |
| | Ctx | 30.14 | 11.10 | 19.00 | 55.55 | | | | | | |
| MH | Con | 38.63 | 15.53 | 24.68 | 62.84 | 2.21 | 3.13 | 49.62 | 67.45 | 64.43 | 31.39 |
| | Ctx | 29.82 | 10.61 | 18.51 | 55.24 | | | | | | |
| MH+INF | Con | 39.43 | 15.75 | 24.77 | 63.11 | 2.26 | 3.13 | 51.56 | 68.57 | 64.97 | 32.35 |
| | Ctx | 29.14 | 10.25 | 18.48 | 54.92 | | | | | | |

## 7 CONCLUSIONS

In this paper, we propose *disentangled paper summarization*, a new task in scientific paper summarizing where models simultaneously generate contribution and context summaries. With the task in mind, we introduced a large-scale dataset with fine-grained reference summaries and rich meta-data. Along with the data, we introduced three abstractive baseline approaches to solving the new task and thoroughly assessed them using a comprehensive evaluation protocol design for the task at hand. Through human studies involving expert annotators with motivated the usefulness of the task in comparison to the current scientific paper summarization setting. Together with this paper, we release the code, trained model checkpoints, and data preprocessing scripts to support future work in this direction. We hope this work will positively contribute to creating AI-based tools for assisting scientists in the research process.

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
