# OpenReview forum: "What's new? Summarizing Contributions in Scientific Literature"
_ICLR.cc/2021/Conference — Reject_

### Official Review · AnonReviewer4 · 2020-10-19
**Interesting direction, needs more work**

**Rating:** 5
**Confidence:** 4

**Review:**

Thanks to the authors for the hard work on this paper. It starts with an interesting and promising premise, and the human evaluation suggests the authors have really hit on something. I have given it a rating of 4 as I feel that this work needs more analysis before it is ready for publication, but I certainly hope to see it published in a more thorough form sometime soon.

The single biggest concern I have is the silver data that is described in "Reference Generation". The authors manually annotated 400 abstracts, and classified each sentence in each abstract into either "contribution-related" or "context-related". They trained a classifier, and obtained 86% accuracy on a held-out dataset. This classifier is then both their source of training data *and* ground truth for everything except the human evaluation. Despite being the foundation of the paper, the section about "Reference Generation" is short and lacks the kind of details I believe it needs for the paper to be truly convincing. Here is a list of some concerns and questions related to this section:

* 86.3% accuracy is not a revealing metric because we don't know how unbalanced the test set is. Please report area under the ROC curve instead.
* I assume you chose 0.5 as the threshold cut-off for the final context/contribution classifier. In Section 5.3 you report that "context summaries include article contribution information." This may be related to the kinds of errors made by the original classifier. What happens when you change the operating point?
* 400 samples for training is relatively small. How representative are they of the different scientific fields in Table 4? Abstracts in biomedicine are very different than abstracts in philosophy. What is the held-out AUC per field?
* What if you add another ~50 samples to the 400 samples in the training data? Does your held-out AUC go up? This would help you assess if 400 documents is enough.
* How much inter-annotator agreement is there for this task? How many annotators did you use per document?
* Let's say the abstract is composed of a series of sentences: ABCDEFG, and the "contributions" are BDFG and "context" are ACE. These are non-contiguous passages, and are likely less fluent than the original abstract. How does this reduction in fluency affect your summarization algorithms? Is there a reduction in fluency in the generated passages as well?

Other questions/concerns:
* In data curation, the number of incoming/outgoing citations was limited to 20. Why?
* If I was to imagine a use for a real system based on this work, it would be applied to academic papers shortly after publication. This means that in a production setting, there will be *no* incoming citations. Does your algorithm still work in this case?
* As far as I understand, you used a distilBART model to summarize the entire full text of the paper, without the abstract, but including the title. Is this correct? If so, please clarify exactly what the input to the model is. How do you deal with long documents using a model that has a much smaller maximum input?
* In Eq (2) - how do you compute $p(.| D)$, $p(.| C_O)$, $p(.| C_I)$?
* It's hard to interpret the numbers in Table 2 without some sort of baseline to compare against. Please consider adding a semi-naive baseline of some sort.
* Please provide a bit of information about how you set the various hyper-parameters in section 5.2. By cross-validation? Hand-tuning?
* In section 6.1 you state that "the factual correctness ... could not be assessed". For scientific text factual correctness of any summarization system is essentially a requirement. Given that you have access to expert evaluation, I think it's critical that some evaluation of factual correctness is done.
* For the human evaluation - how many annotators did you use per document? What was the inter-annotator agreement?

---

> ### Author Response · Authors · 2020-11-19
> **Response to the review**
>
> > Report Area under ROC instead of the accuracy
>
> Thank you very much for your valuable suggestion. The area under the ROC curve was 0.932, we added this information to the paper.
>
> > Change in the threshold of the classifier may affect the summaries' overlapping issue
>
> This is a great suggestion, thank you very much. We picked 0.5 as it resulted in the best accuracy, but changing the value would definitely affect the annotations, thus the trained model outputs. We will conduct this experiment and report the differences in the summaries.
>
> > 400 samples for training is small for a classifier
>
> We would like to highlight that our classifier is operating at sentence-level. Hence, the total number of examples is 3341 as stated in the beginning of page 4 (Section 3.1, last paragraph). Minor domains were missing in the abstracts; we will update the classifier with the better coverage of domains and proper data augmentation if necessary to account for the label imbalance.
>
> > Inter-annotator agreement
>
> Thank you for the question, this was a concern shared by all of the reviewers. We calculated Fleiss’ Kappa for the usefulness evaluation and they were 0.33 and 0.41 for CORD and S2ORC, respectively. This information was added in the revised version of the paper (Section 6.3)
>
> > Potentially non-contiguous passages in the data
>
> This is a valid concern, however, we did not observe significant issues with the fluency of generated outputs. We attribute this to using abstractive summarization models that leverage pre-trained language models. The benefit of using abstractive, rather than extractive, methods is that the models have the ability to paraphrase and fuse disjoint sentences. Using pre-trained language models as the backbone of our baseline models allowed us to transfer the models' general knowledge of language modeling into the tackled task.
>
> > The number of incoming/outgoing citations being 20
>
> This is a hyperparameter of the data generation process and we could easily add more. In the preliminary stage, we varied the number from 5 to 50 and found that the number of papers to parse quickly blew up, so we stayed with 20. However, it is arguable that this selection is arbitrary and we will consider introducing a more systematic cut-off criterion.
>
> > Real system use case of a new paper
>
> Our models require inbound/outbound citations only during training. Trained models can be applied to any papers (new or old), independent of the availability of incoming and outgoing citations, to produce contribution and context summaries. This was shown in our paper in Section 6.3, where the trained baselines were applied to a set of recently released papers related to Covid-19.
>
> > Handling the input to the model
>
> We only feed the main text (i.e., starting from the introduction) to the model. We do not feed the paper title or any other metadata to the model. Handling long inputs is an ongoing challenge within text summarization and other natural language generation tasks, and an active research area in itself. We have explored incorporating efficient attention mechanisms to accommodate full paper text in the preliminary stage. However, this performed worse than a BART-based model with truncated inputs, so we did not pursue this direction further. We believe proper pre-training on longer inputs is a necessary step and leave that as future work.
>
> > Computing the conditional probabilities in Eq 2
>
> Each quantity is calculated by feeding the conditioned information (e.g. $C_O$) as the input to the encoder and target sequence as the input to the decoder (e.g. $y_{con}$) in a teacher-forcing fashion, and then computing the loss values. In short, we do a forward pass of the model for calculating each conditional probability.

---

### Official Review · AnonReviewer3 · 2020-10-25
**Official Blind Review #3**

**Rating:** 4
**Confidence:** 4

**Review:**

This paper discussed the task of generating scientific paper summaries along two axes: contribution (ie. novelty) and context/background. The point is two disentangle these two, since often different readers would be interested in only one of these. While I like the idea of this paper, I think it has the following issues:

- Evaluation: There is no evaluation against a manually annotated gold standard. The references used in section 5 are based on the output of a classifier trained on manual annotations, which itself was not evaluated against manual annotations either. Thus I am rather concerned about what we can learn from the results presented. Section 6 presents some observations in 6.1. but even though they are useful on can't read much in them. And I am not convinced that the lack of expert annotators means that one cannot assess hallucinations; if one can assess whether the novelty is extracted correctly (in the same section), then hallucinations should be possible to assess. In section 6.3 the human evaluation is essentially comparing the model proposed (unclear which of the variants though) against the output of a summarization model that doesn't separate novelty from context. However we don't know how good this summarization model is, let alone that asking about usefulness only conflates a number of aspects that matter in a summary, such as informativeness, fluency, factual correctness, readability, etc. Overall, I think the evaluation is not adequate for a paper at a top conference.

- Modelling: Why not experiment with extractive summarization? Abstracts often overlap with the main paper, and you have a sentence level classifier anyway. And the hallucination wouldn't be an issue then. Also there is previous work on modelling citation function: https://www.cl.cam.ac.uk/~sht25/papers/emnlp06.pdf which should be acknowledged but also could be useful in creating better data and models. Similarly, there is work on scientific article zoning: http://oro.open.ac.uk/58880/

- I am not sure I follow the second branch in equation 2: why should context surprise the articles citing the one being summarized? They would share the context quite often. Also I don't see the connection to the disentanglement.

---

> ### Author Response · Authors · 2020-11-19
> **Response to the review**
>
> > No evaluation against a manually annotated gold standard
>
> This study was previously included in the Appendix, we have moved it to the main content of the paper (Section 6.4). The performance trends were similar to when we use the silver standard.
>
> > Hallucinations should be possible to assess if novelty can be assessed
>
> In our automatic evaluation, the novelty metric simply measures the textual similarity to the reference contribution summary (and the dissimilarity to the context summary). Since this is done at summary-level, it is not straightforward to assess the degree of hallucinations in a fine-grained fashion. We believe a factual consistency evaluation between the source article (full paper) and the contribution summary is critical for text summarization, however, no existing automatic methods are suitable for our purpose and manual annotations are infeasible (please see the response to Reviewer 2).
>
> > Exploring extractive summarization
>
> We agree that extractive summarization models are a viable way of conducting scientific paper summarization. If executed on a sentence-level, they do address most (not all) issues with factual consistency of summaries. However, in our work, we decided to focus on abstractive approaches to the introduced task, due to the benefits coming from the model's ability to paraphrase and fuse disjoint sentences. We appreciate the two related work on fine-grained annotation of scientific papers and encourage future work applying extractive summarization models to the task introduced in our paper.
>
> > Branching in Equation 2
>
> Sorry for the lack of clarity in the derivation of the second branch of Eq 2. Here, $C_I$ is capturing exclusively the information about the contribution of the paper being summarized (let it be $X$) and not the context. This is because inbound citations would only cite $X$ to describe the work provided by $X$, not the background context of $X$. On the contrary, $y_{ctx}$ is the context of $X$. Therefore, content discussed in $C_I$ is less likely discussed in $y_{ctx}$, i.e. surprisal is observed. For example, pre-trained language model-based summarization papers' contribution will less likely discuss pointer-generator network model [1], or even TransformerAbs (Liu et al., 2019).
>
> [1] See, Abigail, Peter J. Liu, and Christopher D. Manning. "Get To The Point: Summarization with Pointer-Generator Networks." Proceedings of the 55th Annual Meeting of the Association for Computational Linguistics (Volume 1: Long Papers). 2017.
> [2] Liu, Yang, and Mirella Lapata. "Text Summarization with Pretrained Encoders." Proceedings of the 2019 Conference on Empirical Methods in Natural Language Processing and the 9th International Joint Conference on Natural Language Processing (EMNLP-IJCNLP). 2019.

---

> > ### Comment · AnonReviewer3 · 2020-11-19
> > **Response to author response**
> >
> > While I appreciate the response, I am afraid that it confirms my concerns with the evaluation. Not being able to evaluate the factual accuracy of summarization focused on the background, compounded with the lack of exploration of methods are more suitable to achieve it (extractive summarization) suggest that this works need to be improved before being published in a top conference.

---

> > > ### Author Response · Authors · 2020-11-20
> > > **Response regarding the factuality evaluation and the use of extractive models**
> > >
> > > We appreciate and share the concern about generating factually correct summaries. However, currently there are no available methods that could be used to reliably evaluate our models' outputs. We believe that offering unreliable and possibly false numbers that would aim to quantify this aspect could cause more harm, in comparison to discussing and highlighting this shortcoming. Also, as mentioned before, developing a robust metric that would evaluate the factual consistency of summaries is a task within Natural Language Processing and is beyond the scope of our work.
> > >
> > > We would like to highlight that the majority of papers on text summarization that are submitted and accepted at top-tier ML and NLP conferences do not provide experiments that evaluate this aspect and they do not provide extractive methods as a solution.
> > >
> > > Please find below two examples of papers that were very well received at ICLR and are considered seminal papers in the field which discuss abstractive models but do not offer factuality evaluation:
> > >
> > > [1] https://openreview.net/forum?id=HkAClQgA-
> > >
> > > [2] https://openreview.net/forum?id=Hyg0vbWC-
> > >
> > > Other examples of more recent work, accepted to top-tier conferences and journals which does not offer such evaluation for abstractive models:
> > >
> > > [3] https://www.aclweb.org/anthology/N18-2097/ (NAACL2018)
> > >
> > > [4] https://icml.cc/virtual/2020/poster/5953 (ICML2020)
> > >
> > > [5] https://papers.nips.cc/paper/2019/hash/c20bb2d9a50d5ac1f713f8b34d9aac5a-Abstract.html (NeurIPS2019)
> > >
> > > [6] https://proceedings.icml.cc/static/paper_files/icml/2020/3934-Paper.pdf (ICML2020)
> > >
> > > [7] https://arxiv.org/abs/1910.10683 (JMLR)
> > >
> > > [3] is the very paper that introduced the scientific document summarization task (ArXiv and PubMed) but neither provides the extractive model as a solution nor evaluates the factual consistency of generated summaries. [4] evaluated their model on the same dataset without reporting factual correctness.
> > >
> > > Using sentence-level extractive methods does not guarantee that generated summaries will be factually correct. It does solve the issue of factuality within the extracted sentence, however, it leaves unchecked factual inconsistencies stemming from relations between sentences, such as temporal dependencies.
> > >
> > > [Edit: clarified about the lack of factuality evaluation on the two ICLR papers mentioned above.]

---

### Official Review · AnonReviewer1 · 2020-10-28
**Confused about the major contribution of this work**

**Rating:** 4
**Confidence:** 5

**Review:**

This work presents a dataset based on S2ORC with additional annotations on the abstracts in terms of contribution and context. Based on this dataset, this paper also adopts two baseline models from prior work with a training strategy (also defined by prior work) to demonstrate the task of summarizing scientific literature from two aspects.

Overall, I was confused about the contribution of this work. Based on my understanding it could either be presenting a newly annotated dataset or demonstrate this task using some baseline models. However, neither of them has enough evidence to convince me that this is a solid work.

About the data, I have a few questions about the annotation procedure:

1) about data curation, if there are more than 20 incoming citations, the proposed method suggests to keep only the top 20 papers based on their citations. However, how do we know these 20 papers are the most relevant to a specific work?

2) I had trouble to understand the value of "silver" standard references. First of all, the classification accuracy is only 86.3%, without further guarantee, we have no idea about the annotation quality. Besides, if the major contribution of this work is about the dataset, then manually annotating 400 abstracts really did not sound like enough contribution.

My confusion about the contribution of this work mainly came from section 5 and 6. Particularly, in section 6, the line "to better understand the strengths and shortcomings of our models" reminds me that maybe the contribution of this work is about those models. Another confusion is that, if this work is about the dataset, then I really think it is necessary to analyze the annotation quality, instead of the performance of existing models.

---

> ### Author Response · Authors · 2020-11-19
> **Response to the review**
>
> Thank you very much for your review! Regarding the raised questions:
>
> > How we know the top 20 papers based on their citations are most relevant to a specific work
>
> We apologize for the unclear description. We found that the mentions regarding the selection of 20 citations on the paper were erroneous and we fixed them in the paper. Specifically, if the number of inbound and outbound citations (each) exceeds 20, we simply sample 20 and sort in descending order by the number of citations made from/to the target paper. Therefore, our paper selection does not guarantee that 20 citations are the most relevant work to the target article. Ideally, one could parse every citation and assess the relevance through similarity measures or using other metadata while being expensive to conduct on all the articles. We aim to improve this part in future work.
>
> > The quality of silver standard references
>
> We have compared the silver standard annotation against the gold standard manual annotation by evaluating the trained models on each set separately and noted their results show similar trends (originally in the Appendix). Based on this, we argue that as a whole, the silver standard annotation can be the surrogate for training and evaluating the model. We moved the relevant section to the main paper’s Section 6.4.
>
> > 400 abstracts are not enough as the contribution if it is about the dataset
>
> Considering the highly specialized language and content of scientific articles, collecting reliable and high-quality manual annotations requires expert annotators and cannot be outsourced to crowd-source workers. This makes it infeasible to obtain large-scale manual annotations due to the limited availability of expert annotators, their time constraints, and high annotation costs. We believe that the manually annotated 400 abstracts (3341 individual sentences) would be sufficient in case evaluation against the gold standard was necessary. Moreover, as mentioned above, we verified that our silver standard annotation exhibits similar evaluation results to those with the gold standard annotation.

---

### Official Review · AnonReviewer2 · 2020-10-29
**Interesting task, shows promise, but needs more extensive experiments**

**Rating:** 5
**Confidence:** 4

**Review:**

This paper introduces disentangled paper representation, which tries to generate paper summaries that explicitly differentiate between the contributions and the context of the work.  I like the general direction and the approaches, but I feel that the experiments and analysis need more work.

Strengths:
I generally think decomposing scientific article summaries into two pieces, contextual descriptions vs. contributions, would be helpful.  The two methods evaluated here (one using control codes, the other using a multi-head architecture) seem reasonable.  Also, the informativeness approach (equation 2) based on inbound vs. outbound citations is well-motivated and interesting.  The paper uses a reasonable set of automated metrics on a silver data source, and I found those experiments to be relatively well-done although ultimately the results are somewhat inconclusive, in that there were not strong differences between different techniques.

Weaknesses:
Because automated metrics are always highly imperfect, human evaluation of the summaries is important.  Unfortunately I think the human evaluation in this paper leaves too many questions unanswered.  First, some details are missing -- what was the inter-annotator agreement, and what exactly was the prompt given to the annotators?  Further, because the experiment tests standard output (regular abstract format) against a newer thing (with context and contribution structure), it seems possible that annotators were biased to prefer the disentangled summaries just because they’ve novel and/or suspected to be the target of the study.  Some discussion of this bias is necessary.  Also, more analysis of why the annotators preferred the disentangled summaries would help alleviate the bias concerns (ideally, the paper would perform a task-based evaluation showing that the disentangled summaries help users more rapidly or accurately browse the literature).

Additionally, the truthfulness of the generation seems like a potential concern that is not discussed.  For example, the contextual output in Table 3 includes a lot of general facts about the domain that are different from the ones shown for the given abstract.  Are all of these statements factual, i.e. from the source documents, or does the abstractive model “make things up”?  The evaluation does not directly address this question.

Suggestions/questions:
I’d like to see an analysis of how important it is to have the inbound citations---these will not exist for the most recent papers, which are arguably the most valuable ones to summarize.  While the paper does ablate the informativeness measure based on the citations it does not experiment with removing the inbound citation text from the input, I would like to see that.

The ablation studies (Table 2) suggest that the informativeness objective does not particularly help for the control codes model, even with disentanglement.  But multi-head is helped more by informativeness (or more precisely, multi-head suffers with out it).  The paper notes this effect, but can you explain more why MH and CC differ in this way?

---

> ### Author Response · Authors · 2020-11-19
> **Response to the review**
>
> We appreciate the detailed review. Below we address the raised concerns:
>
> > Missing inter-annotator agreement and human annotation details
>
> Thank you for pointing this out. For the usefulness evaluation, Fleiss’ Kappa for CORD and S2ORC were 0.33 and 0.41, respectively. The human study was conducted in a hypothetical paper-reviewing setting, where judges were asked to review the target papers and were presented model-generated summaries that they could use for assistance. For each reviewed paper, the annotators were shown two associated summaries, one coming from a traditional paper summarization model which was trained to generate abstracts and the other generated by our model. The judges were asked to assess: “Which of the generated summaries would you consider more helpful for your task (reviewing)?”. Strict measures were taken to ensure that the study was not biased. We added the details of the human study in Section 6.3.
>
> > It is possible that the annotators get biased.
>
> We apologize for the lack of clarity in terms of the possibility of introducing biases into the study. However, we would like to clarify that strict measures were taken to conduct an unbiased study. Specifically, our annotators were: 1) not informed about the objective of the study, 2) not associated with the study in any way, prior to being asked to participate in the evaluation, and 3) not engaged in research related to neural text summarization.
>
> > Truthfulness of the summaries
>
> We agree that factual consistency (or truthfulness) of generated summaries is a crucial aspect of text summarization model evaluation.
> When developing our models we considered different automatic and manual approaches to factual consistency evaluation, however, we could not find an approach that could be reliably applied in our setting. Automatic methods could not be used due to: 1) very long source documents make it impossible to adopt entailment-based methods due to length limits of existing models [2], and 2) the large variety of covered domains makes it impossible to use specialized relation extraction systems to harvest and compare fact triples and domain-agnostic IE methods, such as OpenIE, were reported to have lower correlation with human judgment than ROUGE in terms of factuality [3]. The length of input documents and specialized domains of summarized articles require high expertise and large time investment from human judges making it infeasible to hire experts or crowdsource annotators for the task at hand.
>
> Factual consistency evaluation is a broader problem within text summarization[1], however, developing novel and robust evaluating metrics for factual consistency is beyond the scope of our work.
>
> [1] Kryscinski et al. "Neural Text Summarization: A Critical Evaluation" Proceedings of the 2020 Conference on Empirical Methods in Natural Language Processing (EMNLP). 2020
> [1] Kryscinski et al. "Evaluating the factual consistency of abstractive text summarization." Proceedings of the 2020 Conference on Empirical Methods in Natural Language Processing (EMNLP). 2020.
> [3] Goodrich et al. "Assessing the factual accuracy of generated text." Proceedings of the 25th ACM SIGKDD International Conference on Knowledge Discovery & Data Mining. 2019.
>
>
>
> > The importance of the inbound citations that are not available for the latest papers
>
> We would like to clarify that the inbound and outbound citations are incorporated into the objective functions only at training time. The input to the model at inference time is only the article itself. Thus, the proposed models can be successfully applied to the latest research papers, as presented in our work by summarizing scientific articles related to Covid-19.
>
> > Informativeness objective does not help for CC, but "helps" for MH. Why this is the case.
>
> The two models differ in terms of the degree of parameter sharing between the two generation modes, therefore it is not possible to attribute the difference in performance based on architecture. However, an important difference between the two models is that CC model can condition both the encoder and the decoder with the output mode of interest, while MH model, despite having separate parameters for the two modes, has to decode from the shard encoder representation. We suspect this is partially why MH is worse than CC without the help of the informativeness objective. With the objective, more gradients computed with respect to the right target summaries help fill the gap, as MH model could require more supervision signals to train mode-specific parameters.

---

> > ### Comment · AnonReviewer2 · 2020-11-24
> > **Thank you for the response**
> >
> > I appreciate the response, it addresses some of my concerns.  Thank you for pointing out my misunderstanding regarding the inbound citations.  I'll increase my score to a 5.  I do think that adding manual annotation for factuality (a concern also brought up by other reviewers) would improve the paper substantially.

---

### Decision · Program_Chairs · 2021-01-07
**Final Decision**

**Decision:**

Reject

**Comment:**

The paper attempts at generating two types of summaries for scientific papers: summary of contribution and summary of background context. Most reviewers appreciated the motivation and found this research to be quite interesting and useful, however all reviewers had concerns regarding both execution/presentation of the ideas. While authors try to address some of the concerns, there are many clarifying questions and points raised by reviewers. Addressing all these points requires rather a major revision. Therefore the paper is not quite ready yet and would benefit from another iteration.